# Capturing cross-session neural population variability through self-supervised identification of consistent neuron ensembles

**Justin Jude**                                         JUSTIN.JUDE@ED.AC.UK
*School of Informatics, University of Edinburgh, Edinburgh, Scotland, EH8 9AB*

**Matthew G. Perich**                                   MPERICH@GMAIL.COM
*Université de Montréal and Mila, Montréal, QC, Canada H3C 3J7*

**Lee E. Miller**                                       LM@NORTHWESTERN.EDU
*Feinberg School of Medicine, Northwestern, Chicago, IL 6061*

**Matthias H. Hennig**                                  M.HENNIG@ED.AC.UK
*School of Informatics, University of Edinburgh, Edinburgh, Scotland, EH8 9AB*

**Editors:** Sophia Sanborn, Christian Shewmake, Simone Azeglio, Arianna Di Bernardo, Nina Miolane

## Abstract

Decoding stimuli or behaviour from recorded neural activity is a common approach to interrogate brain function in research, and an essential part of brain-computer and brain-machine interfaces. Reliable decoding even from small neural populations is possible because high dimensional neural population activity typically occupies low dimensional manifolds that are discoverable with suitable latent variable models. Over time however, drifts in activity of individual neurons and instabilities in neural recording devices can be substantial, making stable decoding over days and weeks impractical. While this drift cannot be predicted on an individual neuron level, population level variations over consecutive recording sessions such as differing sets of neurons and varying permutations of consistent neurons in recorded data may be learnable when the underlying manifold is stable over time. Classification of consistent versus unfamiliar neurons across sessions and accounting for deviations in the order of consistent recording neurons across sessions of recordings may then maintain decoding performance and uncover a task-related neural manifold. Here we show that self-supervised training of a deep neural network can be used to compensate for this inter-session variability. As a result, a sequential autoencoding model can maintain state-of-the-art behaviour decoding performance for completely unseen recording sessions several days into the future. Our approach only requires a single recording session for training the model, and is a step towards reliable, recalibration-free brain computer interfaces.

**Keywords:** Manifold learning, Neuroscience, Self-supervised learning, Neural decoding, Neural population activity, Sequential autoencoders, Electrophysiology

## 1. Introduction

Neural decoders require stable neurons in a recorded population in order to accurately predict behaviour such as movement or to allow decoding of stimuli. However, over time instabilities in the recording equipment and drift in neural activity lead to instabilities that prevent re-using a decoder trained on one day for a session recorded on another day (Huber et al., 2012; Ziv et al., 2013; Driscoll et al., 2017). At the same time, neural population activity is highly structured and often confined to low-dimensional manifolds (Cunningham and Byron, 2014) that can be recovered using latent variable modelling approaches

(Hurwitz et al., 2021). Importantly, recent work showed that movement-related latent neural dynamics in population activity from the primate motor cortex is stable and could be recovered over intervals as long as two years (Gallego et al., 2020). This suggests that despite the variability at the level of single neurons, in each session a subset of neurons will remain informative about behaviour. A stable cross-session decoder therefore has to be able to identify these neurons and utilise them for decoding. Therefore, here we focus on identifying known recording neurons in unseen sessions. In particular, we hypothesised that a latent encoding of neural activity can be augmented by information about which neurons were seen during training, and at which position in the input. We show that this is sufficient to decode behaviour (in our case different cued arm movements by a monkey with simultaneous motor cortex recordings) with high accuracy across unseen sessions.

We achieve this with a self-supervised approach through training a recurrent neural network (RNN) to predict original neuron positions following data perturbation in a manner mirroring session to session variability. In essence, the closer our perturbations mimic real inter-session variability (as shown in Figure 1), the higher our behaviour prediction performance on an unseen session. These perturbations include adding spikes to existing neurons from randomly generated neurons, removing spikes from existing neurons, shifting the entire neuron population by a constant amount, slightly shifting neurons in time, replacing neurons with randomly generated neurons and eliminating neurons entirely. This

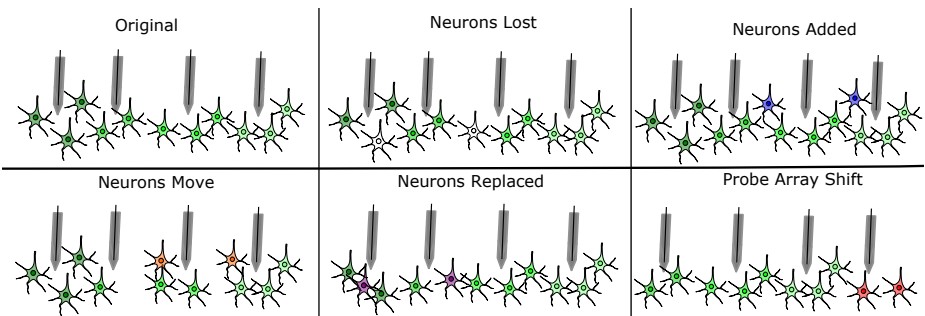

Figure 1: Inter-session ensemble variability possible when recording from neural populations. Neurons from the original recording session can be lost to the recording array, new neurons can become visible, neurons can move between electrodes, original neurons can be replaced by unseen neurons and the entire probe array can shift, causing a systematic change in neuron position. In addition, spike sorting can induce variability as the signal to noise ratio of individual neurons changes between sessions. The perturbations we apply to each trial of recordings is in response to each of these sources of variability. We model each unseen test trial as an instance of a perturbed seen train trial and subsequently, our sequential autoencoder model attempts to map each unseen trial to a known trial.

neuron locator RNN is trained to predict original neuron position within a single recording session from many perturbed variations of trials of this training session. Once trained to predict original neuron positions, a separate network, which in this case is a sequential

autoencoder based on Latent Factor Analysis via Dynamical Systems (LFADS) (Pandarinath et al., 2017), is trained to predict original unperturbed neural recording trials from perturbed variations of trials from the same session. The encoder of this sequential autoencoder receives as additional input the embedding of the neuron locator RNN activations, conditioning the encoder to produce latent variables which are informative enough to accurately reconstruct the original recording. The encoder produces latent variables which are separated by behaviour (arm movement direction) in a self-supervised manner, from which behaviour can be predicted without the model being explicitly trained on behaviour.

Importantly, the joint neuron locator RNN and LFADS encoder ensemble can predict behaviourally relevant latent variables for unseen recording sessions that yield high decoding accuracy. Currently, there are no existing approaches to accurately predict behaviour from an unseen recording session when training on just one single session. We not only show this is possible with our method, but that our approach is robust to inter-session variability for up to 8 days when a sufficient number of neurons are persistent across sessions.

## 2. Related Work

There have been many recent approaches to creating robust behaviour decoders of neural activity (Gallego et al., 2020; Farshchian et al., 2019; Sussillo et al., 2016; Wen et al., 2021; Karpowicz et al., 2022; Wimalasena et al., 2021). However these methods are not capable of decoding behaviour from unseen recording sessions. Recent work in modelling neural activity shows the consequences of selectively perturbing neural data in order to learn relevant latent variables in a self-supervised way using an autoencoder (Liu et al., 2021; Azabou et al., 2021; Zhu et al., 2021). These models take different views of the same neural data and align the latent spaces of these views once passed through an encoder, with the aim of reconstructing these views. We utilise a similar technique to train our sequential autoencoder by aligning the latent variables of perturbed versions of the same data and aim to generate the activity of the original unperturbed trial. Importantly, Liu et al. (2021) propose a model which is invariant to the specific neurons used to represent the neural state within training data; here we look at unseen sessions and do not aim to produce a model invariant to new neurons, but one that is able to identify and utilise seen neurons to reconstruct unperturbed trials.

Gonschorek et al. (2021) and Jude et al. (2022) use domain adaptation to align data across recording sessions. Both use an autoencoder model and a domain classifier. However these models require training on many days of recording sessions. Jude et al. (2022) requires as many as 12 training sessions and training on behaviour explicitly in order to produce high behaviour decoding accuracy on an unseen test session. In this work we achieve state-of-the-art behaviour decoding performance on an unseen test recording session using just one training recording session, and show that this decoding accuracy can be maintained many days into the future without recalibration.

We train an RNN to predict original neuron position from perturbed trials and utilise this network to inform the sequential autoencoder model. Our approach is similar to that used in Noroozi and Favaro (2016), where authors form 9 subsets of images and randomly permute these subsets, then task the model with predicting the permutation.

## 3. M1 Recordings

We apply our model to data from a previously published experiment (Gallego et al., 2020). In this experiment, two monkeys were trained to perform a center-out reach task towards eight outer targets. On a go cue, each monkey moves a manipulandum along a 2D plane to guide a cursor on a screen to the target location. On successful trials a liquid reward is given. Spiking activity from the motor cortex (M1) along with the 2D hand position were recorded during each trial. Spike trains were converted into spike counts in 10ms bins, and behaviour is used at the same resolution. Only successful trials are used, all trials are aligned to movement onset and cut to the shortest reach time across all trials.

For our analysis, we train our model on one session of recorded data from a single day which we denote day 0 (containing 173 trials for both monkeys) and test on subsequent held out days of recordings for each monkey. A comparison of the activity between sessions shows considerable variability, caused by shifts in the order neurons appear as well as disappearance of neurons and the appearance of new ones (see Appendix B, Figure 8). These changes are particularly pronounced for longer time intervals, but are already significant in recordings one day apart. We used 5 days (each with a single session) of recordings for both monkeys, with 55 recorded neurons across all sessions for Monkey C and 17 for Monkey M.

## 4. Data Perturbations

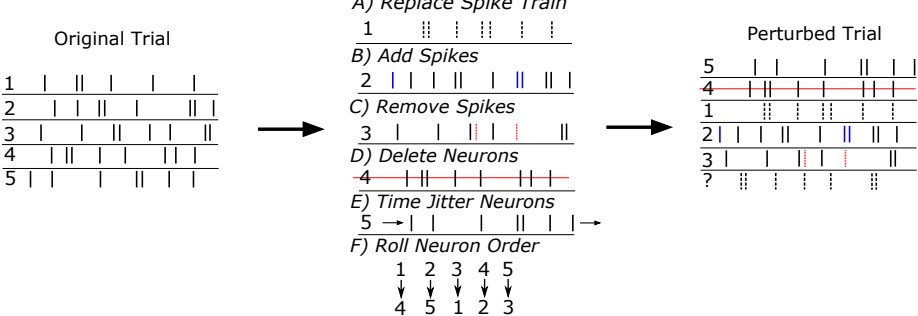

Figure 2: Perturbations applied simultaneously to each trial of neural data, demonstrated with a simple 5 neuron system. A) Replace entire spike train with a randomly generated neuron of the same firing rate as the original neuron. B) Spikes randomly added to spike train proportional to average firing rate of all neurons in a given trial, to mirror influence of nearby newly added unknown neurons. C) Spikes randomly removed to mimic removal or movement of nearby known neurons. D) Deletion of entire neurons to simulate neuron loss between sessions, with randomly generated neurons introduced as the first or last neuron of the trial to keep neuron number consistent. E) Small random time jitter of all neuron spike trains to simulate experimental variation between sessions. F) Constant random shift of the order of all neurons to mirror probe shift.

Fig. 2 outlines the perturbations forming each variation of a single trial during the training of our model. Perturbations A) to D) in Fig. 2 are applied with equal probability to a given neuron of a given trial. Perturbation E) is applied to all neurons, time jitter is chosen randomly between -30ms and +30ms. Perturbation F) is applied to all trials, the amount of this neuron shift is chosen randomly between 0 and 25% of the total number of neurons. We hypothesise that this combination of transformations sufficiently mirrors the real day to day changes of recorded neuron ensembles.

## 5. Model

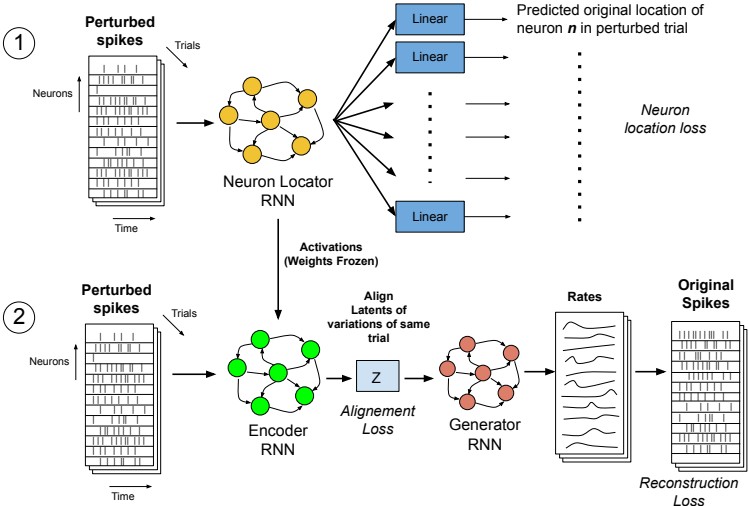

Figure 3: Our model consists of a neuron locator RNN (1) combined with a sequential variational autoencoding approach (2). The neuron locator (1) is trained first to identify original neuron position (or if the neuron is randomly generated) in each trial after perturbations have been applied. Then the neuron locator's weights are frozen and its activations are given as additional input to condition the encoder of the sequential autoencoder (2). We perturb trials when training both the neuron locator and sequential variational autoencoder. The sequential autoencoder is tasked with reconstructing the original unperturbed recording trials. The encoder of the sequential autoencoder maps perturbed versions of the same trial to similar latent variables. We impose an alignment loss across the latent variables of variations of the same trial. The generator RNN of the sequential autoencoder predicts original trials from latent variables produced by the encoder RNN.

Our modelling approach is based on the hypothesis that the perturbations mentioned above can capture the substantial variability between recording sessions from the same animal. We also expect neural activity $x$ is related to the latent variables $z$ through a simple function, however, this function will differ between recording sessions as we expect to observe different neurons in each session. The problem is thus to find the correct encoding

function $z = f(x)$ to transform perturbed neural activity into a consistent latent space which then allows decoding of behaviour. In addition, for the same behaviour we require $z_i$ for each trial $i$ to be similar despite variations in the activity $x_i$. We first train a fully connected layer and an RNN to predict original neuron position in perturbed trials. We apply the perturbations from Figure 2 to each trial, then task the network to predict the original position of each neuron in the recording data or whether it was previously unseen. As shown in Figure 3, for each neuron in the recording data we project a softmax linear read-out layer from the RNN which each form a probability distribution of predicted original neuron position across all possible positions (plus an extra position indicating that the neuron was randomly generated). Each of these is compared against a one hot encoding of the original neuron position before any perturbations have been applied. If the neuron is randomly generated then the one-hot encoding is one at the dedicated extra position. Predictions of original neuron position are made as follows:

$$\bar{x}_{i,1:T} = \mathrm{Perturb}(x_{i,1:T}), \tag{1}$$

$$\mathrm{acts}_i = \mathrm{GRU}_{\theta_{\mathrm{pos}}}(f_{\mathrm{pos}}(\bar{x}_{i,1:T})), \tag{2}$$

$$\mathrm{pos}_{i,n} = \mathrm{softmax}(W^n_{\mathrm{neuron}}.\mathrm{acts}_i) \tag{3}$$

The predicted position for trial $i$ and neuron $n$ is then: $\mathrm{argmax}\, \mathrm{pos}_{i,n}$

Perturb is the application of all perturbations in Section 4 to a given trial. $f_{\mathrm{pos}}$ is a fully connected layer and $\theta_{\mathrm{pos}}$ are the parameters of the locator network used to predict original neuron position. $W^n_{\mathrm{neuron}}$ is the set of linear layers used to predict original neuron position, producing a probability distribution when combined with a softmax layer for each neuron.

Once trained, the weights of this neuron locator network are frozen, and the activations of the RNN are used as additional input to the encoder of an LFADS-inspired sequential autoencoder. This input conditions the encoder in predicting latent variables used to generate original trials from perturbed trials. As proposed by Pandarinath et al. (2017) we assume that the latent dynamics evolve autonomously provided a set of initial conditions $z_i$ that are modelled as Gaussian random variables. These latent variables are produced for each trial by an encoder network consisting of bidirectional Gated Recurrent Units (Cho et al., 2014) (GRU). They are used to reconstruct the original trial-specific neural activity from the perturbed trials. A further bidirectional GRU is used as a generator for neural reconstruction of unperturbed trials from latent variables $z_i$. Training is based on Poisson likelihood for unperturbed neural activity reconstruction (as in (Pandarinath et al., 2017)). The model is trained using real neural activity which corresponds to consistent behaviours. We name our model CAPTure and Identify Variability at Target Ensembles (CAPTIVATE). Implementation details and hyperparameters can be found in Appendix A.

### 5.1. Comparison models

We compare the ability of CAPTIVATE to predict behaviour from sessions of unseen spike data against existing methods and against a variation of our own model (CAPTIVATE-noLoc) where we do not use the locator network trained on original neuron position to aid in aligning perturbed trials. In addition, we look at vanilla LFADS (Pandarinath et al., 2017) in autoencoding trials without any perturbations. We also compare against a baseline RNN

(GRU) with a linear readout layer explicitly trained to reconstruct movement behaviour from neural activity. For all autoencoding models we use a separately trained GRU network to predict behaviour from the day 0 training session latent space.

We do not include ADAN (Farshchian et al., 2019), NoMAD (Karpowicz et al., 2022) or the generative model by Wen et al. (2021) as all require training data from a held out session or subject to be effective. We also do not test against Gonschorek et al. (2021) or Jude et al. (2022) as these approaches require many training sessions to be effective in predicting behaviour from an unseen session whereas we aim to do this with just one training session.

## 6. Results

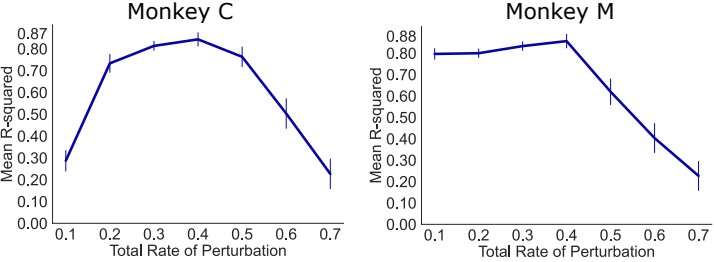

Figure 4: Behaviour decoding performance on an immediately subsequent unseen session (day 1) of CAPTIVATE at different rates of total perturbation. Total perturbation rate is the sum of the rates of perturbations A) - D) outlined in section 4, each of which are applied at equal rates.

Figure 4 shows behaviour decoding performance of CAPTIVATE for an unseen session that was recorded the day after the training session for different total rates of perturbation. A total perturbation rate of 40% (i.e a rate of 10% for each perturbation A) - D) in section 4) for both monkeys appears to be optimal. At perturbation rates above 40%, neural activity from perturbed day 0 train trials with a particular target movement direction begin to resemble original trials of other movement directions, and thus hurt alignment. Perturbation rates below 40%, particularly for Monkey C, are not sufficient to simulate the inter-session variability between day 0 and day 1. Training the neuron locator RNN on a total perturbation rate of 40% for both monkeys yields 85% and 93% accuracy on predicting original neuron position from day 0 perturbed trials from Monkey C and Monkey M respectively. Indeed, the neuron locator network is 76% accurate at identifying original neuron position in a simulated unseen session created with a total perturbation rate of 80% (see Appendix D, Figure 10). Using the optimal rate of 40% of perturbation to trials from both monkeys when training CAPTIVATE leads to the results summarised in Figure 5. For both monkeys we see high behaviour decoding performance on the unseen session from day 1, surpassing previous methods. CAPTIVATE maintains high behaviour decoding performance for Monkey C on an unseen session up to 8 days after the day 0 training session was recorded. CAPTIVATE also accurately maps neurons from trials across unseen sessions of Monkey C up to 8 days into the future to known neurons from trials of the day 0 train

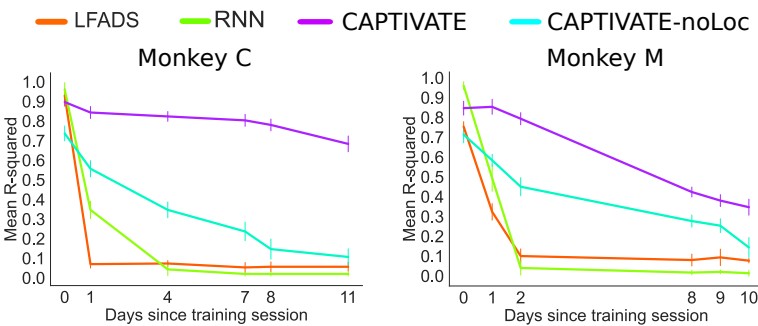

Figure 5: Behaviour prediction performance when testing all models on 30% of held-out trials from day 0 and subsequent days of completely unseen recording sessions. We report the mean $R^2$ between the inferred and true x,y positions. Each model is tested on held out trials from day 0 and trials from unseen sessions recorded an increasing number of days into the future from the original training session (day 0) for both monkeys. Each day 0 train session is run 10 times with different random seeds, with error bars showing standard deviation.

session (see Appendix C, Figure 9). Notably, behaviour decoding for Monkey C is much more stable for future unseen sessions than for Monkey M. This is likely due to sessions from Monkey C containing more than 3 times as many neurons as Monkey M. However, we see in Appendix E, Figure 11 that training CAPTIVATE with 20 neurons from the Monkey C day 0 session is sufficient to achieve an $R^2$ of 0.68 when testing on 20 neurons of the day 8 session, indicating our model can be robust to a low number of neurons.

Notably in the case of Monkey M, day 1 decoding performance is high at all levels of perturbation from 0.1 to 0.4 (Figure 4), therefore it is likely that the session to session variability between day 0 and day 1 is small. Thus, for a subject with fewer neurons in recorded data, CAPTIVATE may only require a low rate of total perturbation when aligning nearby unseen sessions. CAPTIVATE-noLoc, LFADS or an RNN model cannot capture session-to-session variability even for the day 1 unseen session, as shown in Figure 5. CAPTIVATE-noLoc cannot accurately reconstruct original trials from perturbed variations of the day 0 train session, but has a similar day 0 and day 1 session behaviour decoding accuracy, implying our perturbations closely mirror inter-session variability. Therefore poor performance of CAPTIVATE-noLoc is due to the inability of its encoder to recognise known neurons, showing how crucial the neuron locator network is in recognising known neuron ensembles in unseen recordings.

The RNN and LFADS are trained solely on unperturbed trials and so cannot recognise the shifts that occur between sessions. Importantly, none of these models overfit as they yield high decoding accuracies for a held-out portion of day 0 trials for both monkeys and for all models, especially the RNN. Therefore the performance drop of the RNN model when applied to unseen sessions is a clear indication of substantial variations between sessions. Figure 6 shows that for Monkey C, the majority of trials from all unseen sessions are correctly aligned with the corresponding trials in the training data set. The alignment

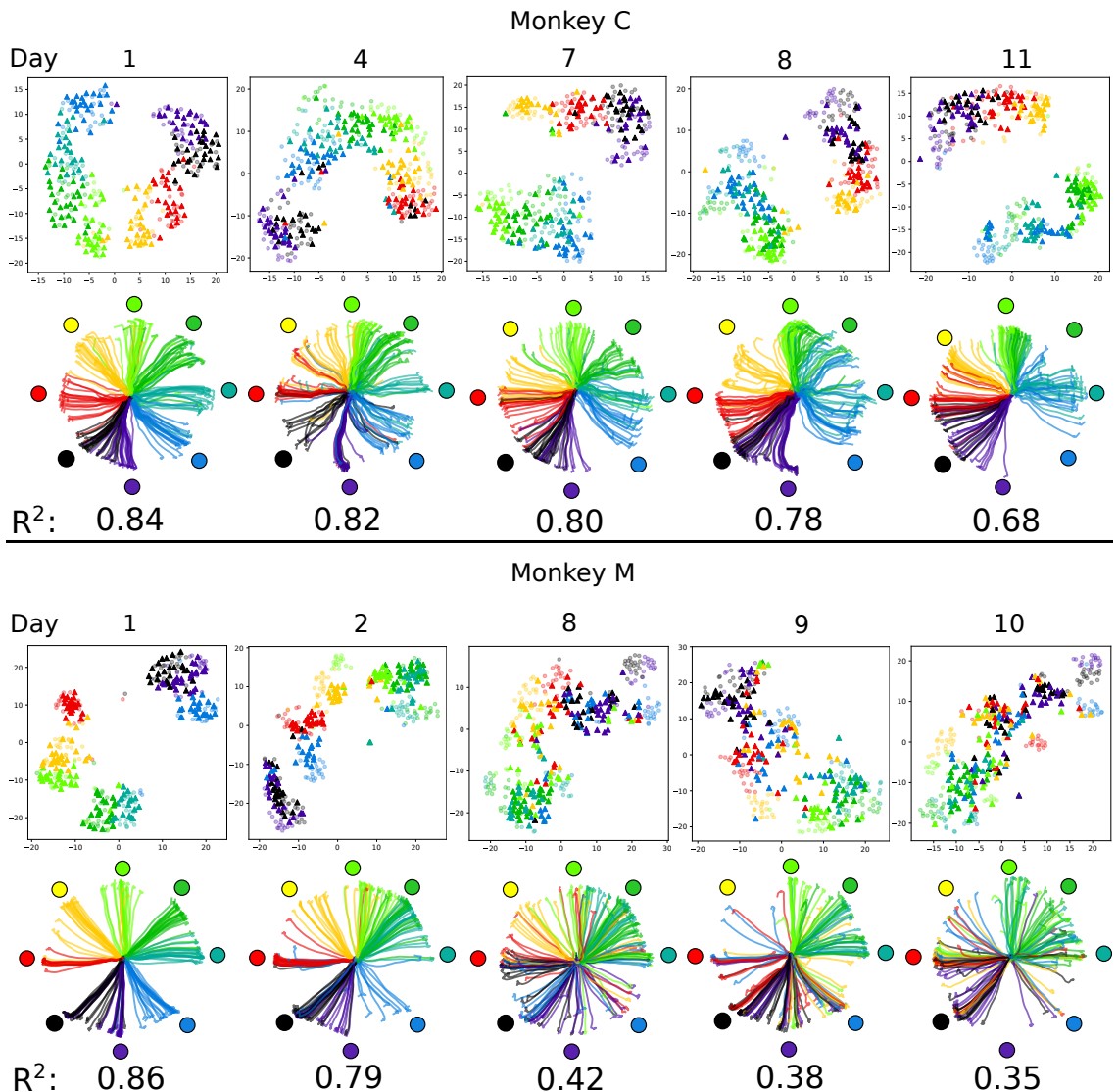

Figure 6: For each monkey, *Top row*: t-SNE embeddings of latent space for CAPTIVATE when applied to each unseen session. In each embedding, points denoted by a circle are trials from the day 0 training session. Points denoted by a triangle are trials from the named unseen session. Each colour represents a target direction for the centre-out reach task. *Bottom row*: Predicted 2D monkey hand position of trials using a separately trained RNN decoder trained only on the day 0 latent space of CAPTIVATE when applied to each unseen session, with mean $R^2$ between all positions of each predicted and ground truth trajectory shown across all trials in a given unseen session.

becomes progressively worse for later sessions, and as the alignment is less precise, behaviour predictions also become worse. Ablations of individual perturbations (outlined in Figure

Table 1: Mean decoding performance effects of ablating individual perturbations when training on day 0 session and tested on day 1 unseen session for both monkeys.

| Ablation | No-Replace | No-Add | No-Remove | No-Delete | No-Jitter | No-Reorder |
|---|---|---|---|---|---|---|
| C Mean $R^2$ | 0.66 | 0.79 | 0.74 | 0.49 | 0.77 | 0.70 |
| | ($\pm$ 0.03) | ($\pm$ 0.01) | ($\pm$ 0.02) | ($\pm$ 0.04) | ($\pm$ 0.01) | ($\pm$ 0.03) |
| M Mean $R^2$ | 0.71 | 0.75 | 0.81 | 0.63 | 0.81 | 0.77 |
| | ($\pm$ 0.03) | ($\pm$ 0.02) | ($\pm$ 0.01) | ($\pm$ 0.03) | ($\pm$ 0.01) | ($\pm$ 0.02) |

2) applied when training on the day 0 session reveal that perturbations which introduce randomly generated neurons and alter the continuous ordering of neurons have the highest impact on unseen session behaviour decoding performance. This analysis is summarised in Table 1 and shows that neuron deletions, replacements and probe shifts cause the majority of inter-session neuron ensemble variability. Additionally, decoding performance across unseen sessions when training on day 0 and day 11 sessions separately is almost symmetrical (Appendix F, Fig. 12), indicating the effective capture of neural variability from unseen sessions both forwards and backwards in time. We further assess robustness by testing CAPTIVATE on a variable number of neurons across sessions (similar to a real BCI setting) and show good generalisation, even surpassing performance of the model trained with 55 neurons (Figure 5) for some unseen sessions (Appendix G, Fig. 13).

## 7. Discussion

We use a self-supervised approach, CAPTIVATE, to train a model to recognise and correct for session-to-session variability in neural recordings. We then show that the combination of this approach with a latent variable model that identifies low-dimensional dynamics in neural activity yields a model that is now robust variability between recordings sessions. The model is capable of successfully predicting behaviour with high accuracy from unseen sessions, surpassing previous work by Jude et al. (2022) when comparing against subsequent day decoding performance. Our approach leads to high and stable behaviour decoding performance on unseen sessions up to 8 days into the future when a sufficient number of neurons are persistent across sessions.

We achieve stable behaviour decoding performance for up to 8 days, which is followed by a slow decline in performance due to an increase in variability that could no longer be compensated. This would require a model to correct even stronger perturbations, but training a model this way leads to an overall decrease in performance even for short time intervals (Figure 4). Therefore long-term stable decoding still requires re-training of the components of a latent variable encoder model such that the altered neural dynamics are re-aligned with the latent dynamics (Karpowicz et al., 2022; Farshchian et al., 2019). Equally, our model fails to successfully decode behaviour from an unseen animal as this requires a more complex mapping function between activity and latent space.

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

## Appendix A. Implementation and training details

The generative process of our model (CAPTIVATE) is:

$$z_i = f_{\text{enc}}(\text{GRU}_{\theta_{\text{enc}}}(\bar{x}_{i,1:T}; \text{acts}_i)), \tag{4}$$

$$g_{1:T} = \text{GRU}_{\theta_{\text{gen}}}(z_i), \tag{5}$$

$$r_t = exp(W_{\text{rate}} . f_{\text{fac}}(g_t)), \tag{6}$$

$$\hat{x}_t \sim \text{Poisson}(r_t) \tag{7}$$

where $\theta_{\text{enc}}$ and $\theta_{\text{gen}}$ are the parameters of the GRUs used to encode perturbed spike trains into latent variables and subsequently generate original unperturbed spike trains from the latent variables. $f_{\text{enc}}$ and $f_{\text{fac}}$ are fully connected layers which produce latent variables and neural activity factors respectively. $W_{\text{rate}}$ is a linear transformation used to generate firing rates at each time step per trial. At each training iteration the following three losses are optimised with Adam (Kingma and Ba, 2015):

$$L_{\text{rec}} = -\sum_{t=1}^{t} \log(\text{Poisson}(x_{i,t}|r_t)) \tag{8}$$

$$L_{kl} = D_{KL}[\text{GRU}_{\theta_{\text{enc}}}(z_i|\bar{x}_i; \text{acts}_i)||\mathcal{N}(0, I)] = -\frac{1}{2}[\log(z_{i,\sigma}^2) - z_{i,\mu}^2 - z_{i,\sigma}^2 + 1] \tag{9}$$

$$L_{\text{align}} = \frac{1}{P} \sum_{j=1}^{p} \sum_{k \neq j}^{p} (z_{i,j} - z_{i,k})^2 \tag{10}$$

Together $L_{\text{rec}}$ and $L_{kl}$ are the usual evidence lower-bound of the marginal log-likelihood in a VAE (Kingma and Welling, 2014). $L_{\text{rec}}$ is minimised by the encoder network and the neural generator network. As in Liu et al. (2021), we apply an alignment loss ($L_{\text{align}}$) across latent variables produced from perturbed trials (where $P$ is the number of perturbations of a given trial) of the same original trial $z_i$ which reduces training duration. We form 2 perturbed variations of each trial in a given batch at each training iteration. Kullback–Leibler ($L_{kl}$) divergence loss (between a multivariate standard Gaussian distribution and the encoder-generated latent variables) and $L_{align}$ are minimised by just the encoder network.

Below are implementation details and hyperparameters for the CAPTIVATE model.

| CAPTIVATE | | |
|---|---|---|
| Parameter | Value | Notes |
| Neuron Locator Network | | Layer Normalisation on all layers |
|   - RNN Units | 784 X 3 | Stacked Gated Recurrent Unit |
|   - $W_{pos}$ Units | 1024 X 3 | Non-linear layer |
|   - $W_{pos}$ Dropout | 0.5 | |
|   - $W_{pos}$ L2 Regularisation | 100.0 | |
| Sequential Autoencoder Encoder | | |
|   - RNN Units | 784 X 3 | Stacked Gated Recurrent Unit |
|   - RNN L2 Kernel Regularisation | 0.1 | |
|   - RNN L2 Recurrent Regularisation | 0.1 | |
|   - $W_{enc}$ Units | 1024 X 3 | Non-linear layer |
|   - $W_{enc}$ L2 Regularisation | 0.1 | |
|   - Latent space dimension | 64 | |
| Sequential Autoencoder Generator | | |
|   - RNN Units | 512 X 3 | Stacked Gated Recurrent Unit |
|   - RNN L2 Kernel Regularisation | 1.0 | |
|   - RNN L2 Recurrent Regularisation | 1.0 | |
|   - $W_{fac}$ Units | 512 | Non-linear layer |
| Training | | |
|   - KL divergence weighting ($\lambda_{kl}$) | 0.02 to 1.0 | Rising exponentially |
|   - Batch size (Train Neuron Locator) | 16 | |
|   - Batch size (Train Seq. Autoencoder) | 4 | |
|   - Learning rate (Train Neuron Locator) | 0.0001 | Adam Optimizer |
|   - Learning rate (Train Seq. Autoencoder) | 0.00001 | Adam Optimizer |

## Appendix B. Changes in recorded neural activity across sessions

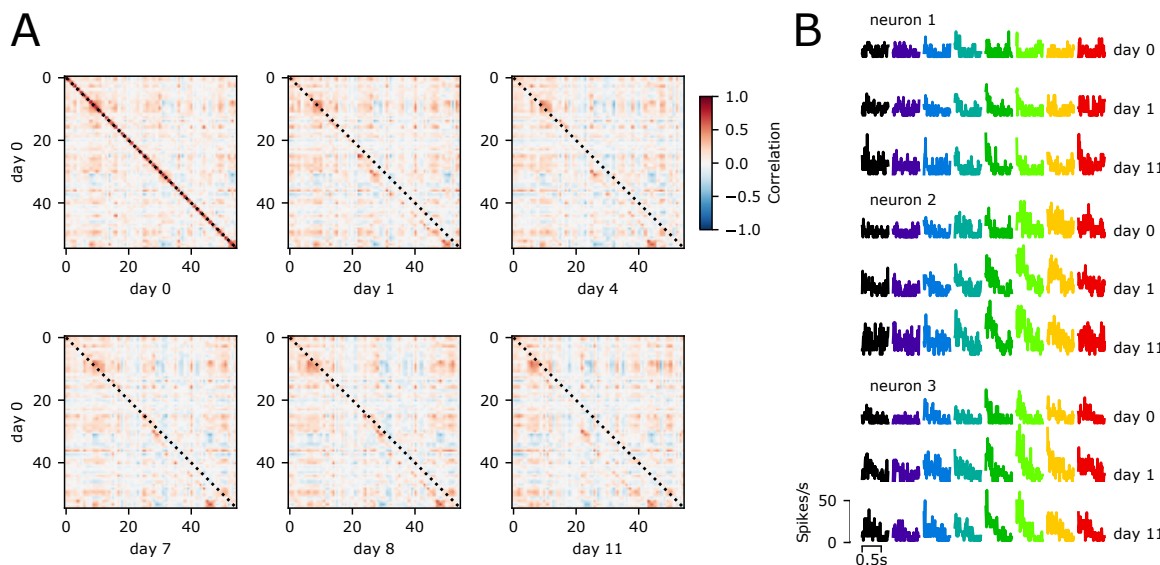

Figure 7: **A**, Pairwise correlations of trial-averaged activity of single neurons between two sessions. For each neuron, the average firing rate was computed for each of the eight movement directions (see part B for examples) at 10ms resolution. The activity for the eight movement directions was concatenated and the Pearson correlation coefficients computed between all neuron pairs. Each plot shows the correlation matrix for activity from session from a different day and activity from the first day (day zero, the training data set in Figure 5). This analysis shows that some neurons from the first session can be matched to neurons recorded at subsequent days, but the relative position of these matched neurons in the recording tends to shift (see high off-diagonal correlations). As the average correlations do not change systematically over this period of time (not illustrated), the gradual changes in neuron identity is a main factor that prevents reliable decoding from unseen sessions in previous models. **B**, Examples of trial-averaged firing rates of three neurons that were tracked over all recording sessions. This matching is based on the similarity of the firing rates, experimentally it is hard to determine if these are indeed the same neurons. In all cases, the time course of the activity is similar and shows consistent differences between trial type (indicated by colour) across sessions. Also note that while these neurons appear to reliably encode movement direction, the activity of a single neuron alone is too noisy to allow for reliable direction decoding from single trials, instead a population decoding approach is required. All data illustrated here is from Monkey C.

## Appendix C. CAPTIVATE accurately maps perturbed neurons and neurons from unseen sessions to known neurons from the Day 0 training session

CAPTIVATE is trained by mapping perturbed trials to known trials. If trials from unseen sessions are similar to the perturbed trials then generalisation to these sessions is possible. Therefore, we aim for the encoder network of CAPTIVATE to map perturbed trials and trials from unseen sessions to day 0 trials. This entails that neurons across unseen sessions (even after neural drift and ensemble change) are mapped directly to neuron positions of the day 0 session at the session. For trials of each movement direction from unseen sessions, we expect that the trial average firing rates of these neurons will map to the day 0 average firing rates for each neuron. As seen below for 4 neurons across 3 sessions (2 unseen), the CAPTIVATE generator network produces trial average firing rates matching the day 0 train session firing rates.

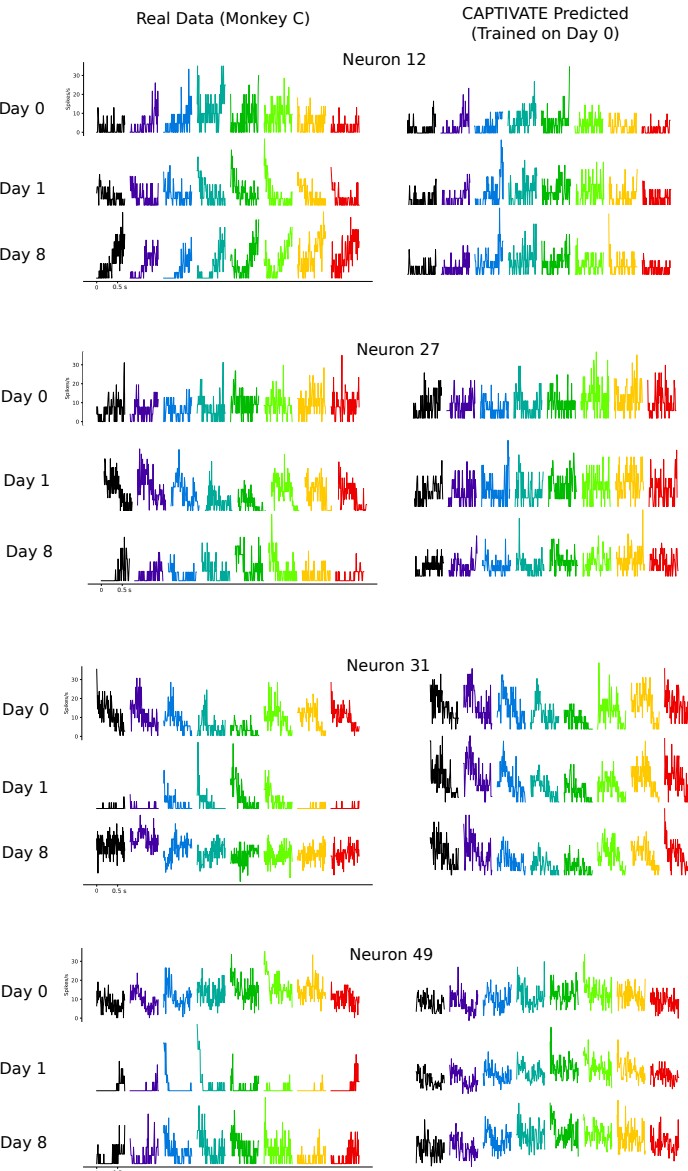

Figure 8: CAPTIVATE is trained on the day 0 session of Monkey C. On the left we show real trial averaged firing rates for each movement direction across 4 randomly selected neurons across the day 0 session and 2 unseen sessions. On the right we show predictions from the generator network of CAPTIVATE. If generalisation is achieved the generator should accurately map neurons across unseen sessions to the neurons of day 0. We see that this is the case as the predicted firing rates are closely matched in the unseen sessions to the day 0 firing rates across movement directions.

## Appendix D. Neuron Locator performance over simulated neural variation

As we do not have ground truth neuron identities from unseen sessions (with respect to the day 0 train session), we simulate inter-session variability by increasing perturbation rate and testing against CAPTIVATE trained on the day 0 session from monkey C with a total perturbation rate of 0.4 (as in the results shown in Figure 6). We see that the neuron locator network of CAPTIVATE can predict neuron identity with 68% accuracy even at a very high total perturbation rate of 1.0.

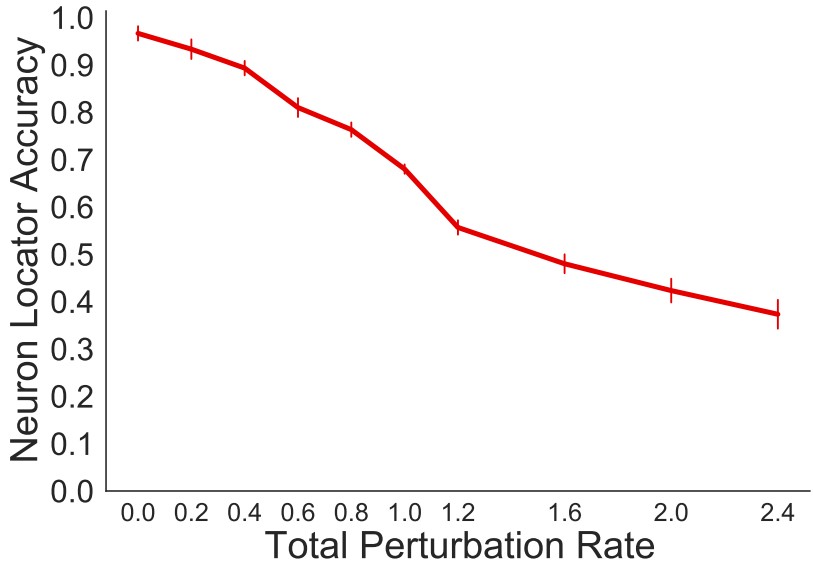

Figure 9: Neuron locator network accuracy when predicting neuron identity (with respect to unperturbed day 0 monkey C train session) as the total rate of perturbation is increased. We are simulating neural drift and ensemble shift across sessions. As we know the ground truth neuron identities, we can assess how well the neuron locator can predict neuron identity.

## Appendix E.  Training and testing CAPTIVATE with different numbers of original neurons

Here we test the varying numbers of neurons across sessions of Monkey C when using CAPTIVATE. We see that only 20 neurons are required across sessions for good generalisation for up to 8 days.

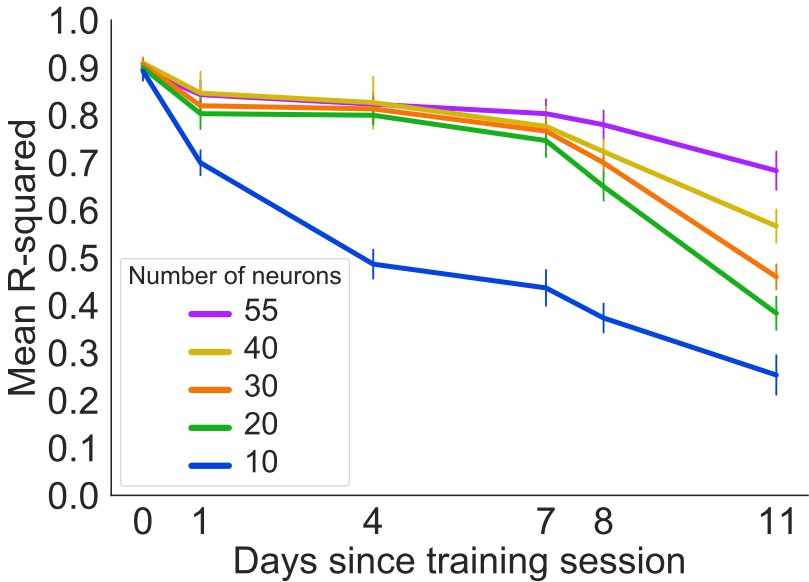

Figure 10:  Behaviour prediction performance when training CAPTIVATE on varying numbers of neurons of the day 0 session recorded from Monkey C and testing on all other unseen sessions of monkey C, using the same number of neurons as used in the training session. We also test all neuron number variations of CAPTIVATE on a held out portion of trials from day 0. We report the mean $R^2$ between the inferred and true x,y positions for the entire movement trajectory of each trial. Each day 0 train session is run 10 times with different random seeds, with error bars showing standard deviation when applied to each unseen session.

## Appendix F.  Changing calibration session

Here we show that by training our model on perturbed trials we can generalise to neural drift and recording array movement. CAPTIVATE accounts not only for session variability in the future but also in the past.

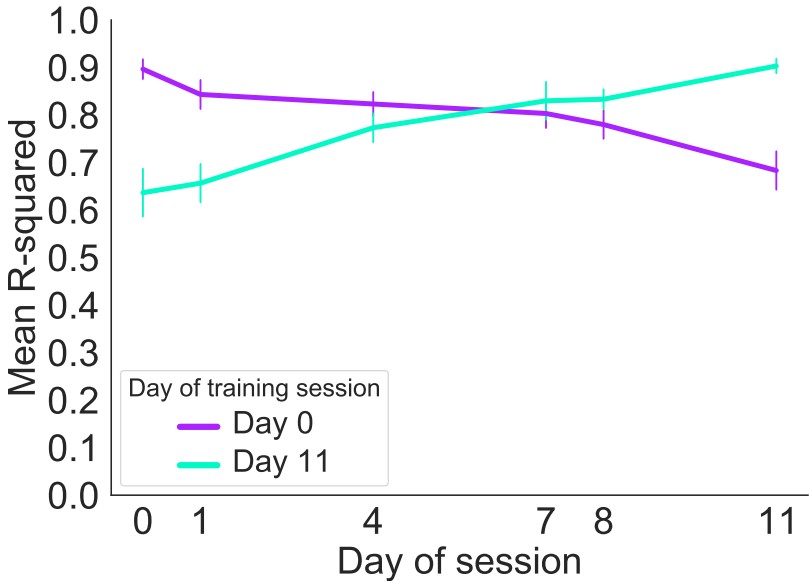

Figure 11:  Behaviour prediction performance when separately training CAPTIVATE on the day 0 and day 11 sessions of Monkey C and testing on all other unseen sessions. We see that performance across unseen sessions when training on these sessions is almost symmetrical, indicating that our model can effectively capture neural variability from sessions both backwards and forwards in time.

## Appendix G. Variable neuron number per session

When using an implanted recording array we may lose electrodes or neurons due to spike sorting error over a period of time. Here we show our model can account for this variable neuron number.

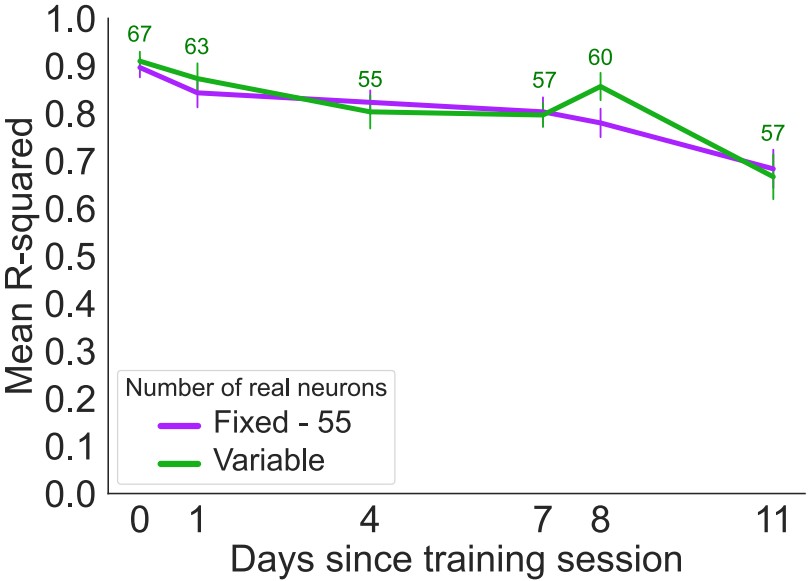

Figure 12: We test CAPTIVATE with a variable number of neurons per session of recording from Monkey C. In our original experiment we only utilise the first 55 neurons of each recording session as this is the minimum number across all sessions. Here we use every neuron available per session (number of neurons per session shown in Figure) and train CAPTIVATE on the day 0 session with 67 neurons. For all other sessions we add randomly generated neurons to compensate. We see that CAPTIVATE is robust to the number of original neurons being variable across sessions. Note the increase in generalisation performance when the model is applied to the day 8 session. This is due to this session having a relatively high number (60) of original neurons, and is thus easier for the model to map trials from this session to known trials from the day 0 training session than it is from other later unseen sessions.

## Appendix H.  Testing trained model on known neural variability

We test the resilience of our whole model against an increasing total rate of perturbation in order to ascertain how much variability the model can account for. For the results below, CAPTIVATE is trained with a total perturbation rate of 0.4 on the day 0 session of Monkey C. Note that the model is trained to map perturbed trials to original unperturbed trials.

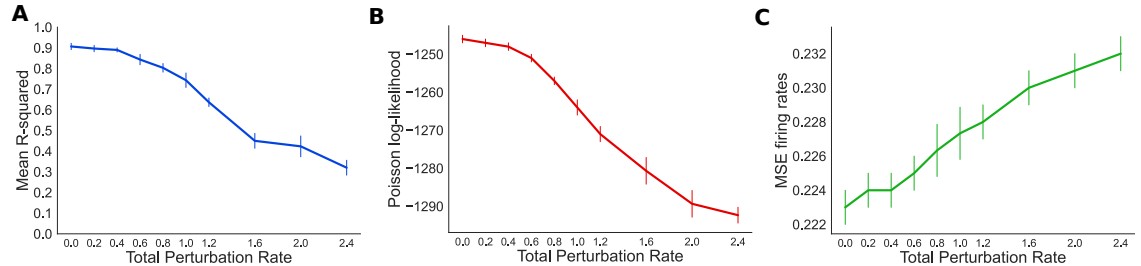

Figure 13: We train CAPTIVATE on trials from the day 0 session of Monkey C with a 0.4 total rate of perturbation on each trial. We then test the trained model on trials of the same session but with increasing rates of total perturbation applied to trials. A) Mean r-squared error of movement predicted from the latent space of the model vs. real movement trajectory of each trial. B) Mean Poisson log-likelihood for neural activity reconstruction by the model generator of original day 0 unperturbed trials. C) Mean squared error of model predicted firing rates vs. real firing rates of original unperturbed day 0 trials.

## Appendix I. Testing trained model on known neural variability across held-out trials

We test the resilience of our whole model against an increasing total rate of perturbation in order to ascertain how much variability the model can account for. For the results below, CAPTIVATE is trained with a total perturbation rate of 0.4 on 70% of the trials of the day 0 session of Monkey C. We show test performance on 30% of the trials of the day 0 session which are withheld from training. Note that the model is trained to map perturbed trials to original unperturbed trials.

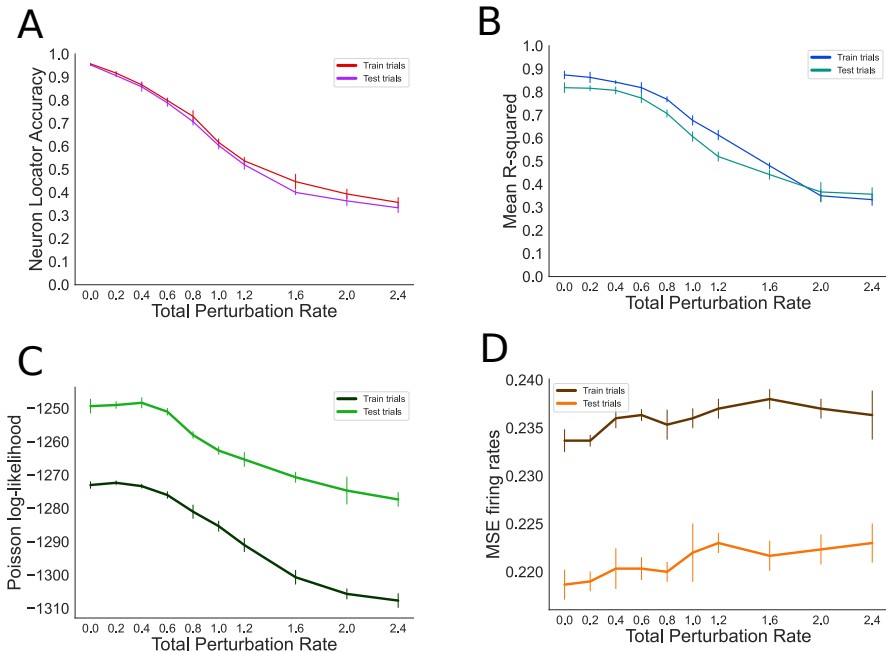

Figure 14: We train CAPTIVATE on 70% of the trials from the day 0 session of Monkey C with a 0.4 total rate of perturbation on each trial. We then test the trained model on the remaining 30% of trials of the same session but with increasing rates of total perturbation applied to these held-out trials. A) Neuron locator network accuracy when predicting neuron identity (with respect to unperturbed day 0 Monkey C train session) as the total rate of perturbation is increased. We are simulating neural drift and ensemble shift across sessions. As we know the ground truth neuron identities, we can assess how well the neuron locator can predict neuron identity. B) Mean r-squared error of movement predicted from the latent space of the model vs. real movement trajectory of each trial. C) Mean Poisson log-likelihood for neural activity reconstruction by the model generator of original day 0 unperturbed trials. D) Mean squared error of model predicted firing rates vs. real firing rates of original unperturbed day 0 trials.

