# OpenReview forum: "Capturing cross-session neural population variability through self-supervised identification of consistent neuron ensembles"
_NeurIPS.cc/2022/Workshop/NeurReps — NeurReps 2022 Poster_

### Official Review · Reviewer_tC3Z · 2022-10-10
**Solid idea for understanding sources of interexperimental variability**

**Confidence:** 4
**Soundness:** 3
**Presentation:** 3
**Contribution:** 3
**Overall Rating:** 6

**Summary:**

The paper proposes that much interexperimental variability is due to loosing, shifts, replacing Neurons etc. They build a model which estimates and reverses these manipulations. This performs well in extrapolating to new unseen recording sessions.

**Questions:**

Can you add the model sizes in Fig 5? Also, how did you ensure that the comparison methods received the same amount of hyper parameter tuning as your method?

In Fig 8 some Neurons seem to be changing their tuning radically across days (ramp up turns to ramp down), and then in the captivate reconstruction this becomes a flattened line. That looks like something more intricate is happening (e.g., drift)?

**Limitations:**

The model cannot properly capture experimental drift beyond the proposed manipulations (see above), pleaser tone down claims.

**Recommended Decision:**

3: Accept

**Relevance:**

2: Limited relevance

**Strengths And Weaknesses:**

The proposed manipulations are quite thoughtful. Having an explicit model for possible mechanistic changes between recording sessions is excellent and more insightful than doing everything with blackbox ML models. It is encouraging that this recovers some of the inter-session variability.

It is a little misleading to discuss neural drift in the introduction. That is a separate issue. The method in this paper can recover some inter-session variability. However, it cannot model changing tuning functions are, e.g., rotations of subspaces in the population responses. This limits the applicability of the method. For instance, this model is not useful if we record very different Neurons in the same animal or from a new animal. Nevertheless, I see much value in coming up with such a constraint useful mechanistic model that can disentangle the effects from changing recording sites and real representational drift.

**Submission Track:**

Proceedings Paper (9 Page)

---

### Official Review · Reviewer_2G8q · 2022-10-11
**Review of “Capturing cross-session neural population variability using self-supervised identification of consistent neural ensembles”**

**Confidence:** 4
**Soundness:** 3
**Presentation:** 3
**Contribution:** 3
**Overall Rating:** 5

**Summary:**

This work proposes an LFADS-based method for robustly decoding behavior from unstable multielectrode recordings of neural activity. Their decorder is trained on data augmented with perturbations that model simple forms of cross-session variability. Using previously-published recordings of motor cortical activity in macaques performing a reaching task, they show that the proposed method, when trained on a single session, can achieve robust decoding across recordings from the ten subsequent days.

**Questions:**

1. In the introduction, the authors write that “Importantly, recent work showed that movement-related latent neural dynamics in population activity from the primate motor cortex is stable […]. This suggests that despite the variability at the level of single neurons, in each session a subset of neurons will remain informative about behaviour.” Is this necessarily true? An alternative possibility is that representational drift is coordinated across the population to maintain a stable subspace that is not aligned to single-neuron axes. This possibility is discussed, for instance, in Rule et al. (Current Opinion in Neurobiology, 2019) or in Rule et al. (eLife 2020).
2. More broadly, the proposed method is, to the best of my understanding, predicated on the assumption that observed drift in raw recording data results from instability in recording methods rather than true variation in neural activity. However, it is possible that the proposed data augmentation could make the decoder more robust to underlying single-trial variability in neural activity or certain forms of longer-timescale drift (e.g., gradual drift in response timing). Could the authors comment on this possibility?
3. The types of recording variability the authors seek to model in their data perturbations are tailored specifically to multielectrode array recordings. In my opinion, it could be useful for the authors to comment on potential (in-) applicability of this method to other long-term recording methods.
4. Unless I am mistaken, the perturbations are applied at the level of spike trains before the conversion to spike counts in 10-ms bins. Does this afford any advantages over perturbing after binning? Moreover, does the binning window appreciably affect results? In a similar vein, how was the +/- 30 ms window for time jittering chosen?
5. The authors only present results for networks trained using the full set of proposed perturbations or with a single perturbation removed. Among single-perturbation ablations, removing neuron deletions incurred the greatest reduction in $R^2$. However, I don’t think this analysis is sufficient to claim that “deletions, replacements, and probe shifts cause the majority of inter-session neuron ensemble variability.” Why not also test which single perturbation provides the greatest gain in decoding robustness relative to a model with no augmentations? A more careful test of which perturbations afford the greatest gain in decoding robustness would be a valuable addition to the paper.

**Limitations:**

In the current version of the manuscript, the authors do not provide extensive discussion of the limitations of their work. I have included what I view as its main limitations among my questions above.

**Recommended Decision:**

2: Borderline

**Relevance:**

2: Limited relevance

**Strengths And Weaknesses:**

In recent years, there has been substantial interest in the stability - or possible lack thereof - of neural activity in premotor areas that drive stable behaviors. An important challenge in experimentally probing the stability of neural representations of stable tasks is inter- and intra-session variability due to recording techniques rather than true drifts in neural activity, as reviewed recently by Driscoll et al. (Current Opinion in Neurobiology, 2022) and by Masset et al. (Biological Cybernetics, 2022). Thus, I think that methods for robust behavioral decoding should be of interest to the community of researchers working on stability and variability in premotor areas. The presentation of the method and the results are generally clear, and the figures are easy to understand. I describe what I view as the weaknesses of this work in my questions for the authors below.

**Submission Track:**

Proceedings Paper (9 Page)

---

### Official Review · Reviewer_fm8h · 2022-10-17
**Using sequential autoencoders to address neural population variability across recording sessions**

**Confidence:** 4
**Soundness:** 3
**Presentation:** 3
**Contribution:** 3
**Overall Rating:** 6

**Summary:**

The authors present a method for decoding on neural population ensembles despite inter-session variability. More specifically, they train a sequential autoencoder to reconstruct perturbed spike trains and characterize decoding performance on a motor task.

**Questions:**

- It would be helpful to include in this paper a short bit about how spike sorting is performed for this dataset.
- The term "perturbation rate" is somewhat confusing given that the individual rates that appear to compose it seem to be probabilities.
- How well would the "baseline" RNN perform (in terms of R^2) on a validation (held-out) set not on Day 0? I would be expect it to be close to .9, given other results, but would be helpful to comment on this.
- I wish there were additional analysis on interpreting the output of the Neuron Locator RNN when analyzed on different days. For example, what percentage of neurons were classified as non-random, and was this similar to the percentage during training? Some visualizations (even in appendix) could be helpful to show the RNN is giving good results.

**Limitations:**

There is no section explicitly addressing the limitations of this work. It would be helpful to get a better understanding of how sensitive the autoencoder/decoding performance is on specific hyperparameters are in this network.

Another point that it would be useful to address is what additional datasets (if any) might be possible to consider in future work, to validate the techniques used here as a general method.

**Recommended Decision:**

3: Accept

**Relevance:**

3: Solid fit

**Strengths And Weaknesses:**

Strengths:
- Nice performance on a difficult dataset, extensive tests of decoding performance across different days, good baselines and additional experiments in appendix
- Addresses an important and challenging problem of interest to computational neuroscience

Weaknesses:
- I wish there were better explanation/motivation in the paper of the choice of architecture being used. The authors do a good job of motivating the problem, referencing previous attempts at similar problems, but the justification for the specific setup appears somewhat ad hoc. Perhaps it isn't, but this could be better explained, perhaps by intuitively laying out how the architecture addresses some of the perturbations added in the model.


**Submission Track:**

Proceedings Paper (9 Page)

---

### Decision · Program_Chairs · 2022-10-21

Accept (Poster)